# Viral and Host Factors Regulating HIV-1 Envelope Protein Trafficking and Particle Incorporation

**DOI:** 10.3390/v14081729

**Published:** 2022-08-05

**Authors:** Boris Anokhin, Paul Spearman

**Affiliations:** Division of Infectious Diseases, Cincinnati Children′s Hospital Medical Center and University of Cincinnati, 3333 Burnet Avenue, Cincinnati, OH 45229, USA

**Keywords:** envelope glycoprotein, Gag protein, recycling, endocytosis, AP-2 complex, AP-1 complex, Rab11-FIP1C, retromer

## Abstract

The HIV-1 envelope glycoprotein (Env) is an essential structural component of the virus, serving as the receptor-binding protein and principal neutralizing determinant. Env trimers are incorporated into developing particles at the plasma membrane of infected cells. Incorporation of HIV-1 Env into particles in T cells and macrophages is regulated by the long Env cytoplasmic tail (CT) and the matrix region of Gag. The CT incorporates motifs that interact with cellular factors involved in endosomal trafficking. Env follows an unusual pathway to arrive at the site of particle assembly, first traversing the secretory pathway to the plasma membrane (PM), then undergoing endocytosis, followed by directed sorting to the site of particle assembly on the PM. Many aspects of Env trafficking remain to be defined, including the sequential events that occur following endocytosis, leading to productive recycling and particle incorporation. This review focuses on the host factors and pathways involved in Env trafficking, and discusses leading models of Env incorporation into particles.

## 1. Introduction

The HIV-1 envelope glycoprotein (Env) directs the process of receptor and coreceptor engagement, leading to fusion and entry into target cells. Env is the major target for neutralization of HIV-1, and as such, it has been a primary focus of vaccine efforts from the early days of the HIV pandemic to the present time. The structure of Env has been intensively investigated, with most studies focused on the ectodomain. However, structural studies of the long cytoplasmic tail (CT) of HIV-1 Env are now available, and will be invaluable for defining interactions of the CT with host trafficking pathways [1,2,3]. An unusual feature of HIV-1 biology is the sparse incorporation of Env trimers on the surface of HIV-1 particles. Estimates of trimer incorporation derived from biochemical and cryoelectron tomography (cryoET) studies range from 7–14 per virion [4,5,6], a very low number compared to viruses such as influenza (300–400 trimers per virion) [7], Ebola (variable with virion length, dense on surface), and measles or respiratory syncytial virus (densely packed on lipid bilayer of virions) [8,9], and somewhat less than SARS-CoV-2 (24 ± 9 spikes) [10]. While it has been postulated that the low number of spikes is a means of immune evasion, the viral and cellular determinants of this low spike density remain uncertain. This is just one of many intriguing aspects of Env incorporation into particles. Another poorly understood and likely related aspect of Env trafficking and particle incorporation stems from the finding that upon reaching the cell surface, Env trimers are rapidly endocytosed [11,12]. Why would a virus that assembles its particles on the plasma membrane (PM) benefit from the removal of its spikes from the PM? Adding to the mysterious nature of Env trafficking and particle incorporation is the fact that the long CT is not required for incorporation into particles within some cell types, while it is required for incorporation (and for replication) in other cell types [13]. 

This review provides an overview of HIV-1 envelope protein trafficking, with a particular focus on host factors and pathways that have been shown to play a role in particle incorporation of Env. We include lessons from other retroviruses, where they may be instructive. We will first provide an overview of HIV-1 assembly and discuss potential models for Env incorporation. We will next describe the synthesis and trafficking of Env through the secretory pathway to the PM, followed by the pathways involved in Env endocytosis, intracellular trafficking, and recycling to the PM. Host factors that interact with Env along trafficking pathways or act to impede its incorporation into particles provide important insights into the process, and will be discussed. Finally, we propose models for Env incorporation based on existing data, and conclude with a discussion of areas for future investigation. 

## 2. HIV-1 Assembly Overview

The central orchestrator of the assembly process for HIV-1 is the precursor polyprotein Pr55^Gag^ (Gag). Gag is synthesized on free cytoplasmic ribosomes, and travels to the plasma membrane (PM) by an incompletely understood route. The domains of Gag represented by proteolytic cleavage products from N- to C-terminus are matrix (MA), capsid (CA), nucleocapsid (NC), and p6, with two small spacer peptides between CA and NC (SP1), and between NC and p6 (SP2). The MA domain plays an important role in the interactions of Gag with the PM. MA is N-terminally myristylated, with myristic acid playing an essential role in membrane interactions and targeting to the PM [14,15]. Gag-Pol is a 160 kD precursor protein that is formed via a ribosomal frameshift near the end of the p6 open reading frame. This frameshift occurs during approximately 5% of the translation events, thus creating a 20:1 ratio of Gag-to-Gag-Pol in the cell [16]. Gag interacts with Gag-Pol, and together they travel to the PM. Trafficking of Gag-Pol to the assembly site brings the viral enzymatic machinery (reverse transcriptase [RT], integrase [IN], and protease [PR] into the developing virions. During and shortly following budding, dimerization of PR leads to cleavage of Gag and Gag-Pol precursors into their component subunit proteins. The cleavage of Gag is an orderly and sequential process. Complete cleavage of Gag is essential for maturation, generating infectious particles. The final step in sequential cleavage of Gag removes SP1 from the C-terminus of CA, and this cleavage event serves as a critical conformational switch in the transition from the immature, spherical Gag shell to the mature, conical viral capsid [17]. 

The viral genomic RNA (gRNA) serves both as a template for the synthesis of Gag and Gag-Pol, as well as the genome for packaging into nascent virions. Gag interacts with the RNA packaging or “psi” signal, a highly structured region of 150–250 bases near the 5′ end of the gRNA [18]. The NC region of Gag plays a critical role in RNA packaging, interacting with this highly structured region of RNA, and incorporating only those gRNAs that have formed a dimer [19]. Encapsidation of two copies of gRNA is essential for viral recombination, as the recombination process has been shown to be required for efficient viral replication and genome integrity [20]. 

The assembly of HIV-1 virions takes place on the PM in model epithelial cell lines and infected T cells. The Gag lattice on the PM can be visualized by electron microscopy as densities underlying the lipid bilayer that sequentially induce membrane curvature, creating a spherical shell that buds from the membrane and is released by a final scission event mediated by the host ESCRT (endosomal sorting complex required for transport) complex [21,22]. Visualization of assembly events at the PM is rare in HIV-infected macrophages, where virions accumulate in a membrane-bound compartment with connections to the PM, termed the virus-containing compartment or VCC [23,24,25]. The VCC may be a site of particle assembly, or alternatively, may be a compartment into which virions captured by tetherin or Siglec-1 at the PM are directed via an endocytic pathway [26,27].

Env is translated on ER-bound ribosomes as a gp160 precursor protein, and forms trimers in the ER. Glycosylation of gp160 trimers is also initiated in the ER. Cleavage to transmembrane (TM, gp41) and surface (SU, gp120) subunits occurs in the Golgi apparatus, along with glycan modifications, and the trimeric complex is delivered to the PM. As already described, Env trimers are then rapidly endocytosed from the PM into early endosomes. Following endocytosis, multiple trafficking pathways are available for Env, leading to productive incorporation into developing particles through recycling pathways, or to degradation in the lysosome. The pathways and host factors involved in the intracellular trafficking of Env are the subject of a more detailed discussion below. The basic aspects of HIV-1 assembly described here are depicted in schematic fashion in Figure 1.

## 3. Cell Type-Specific Incorporation of Env and Models for Env Incorporation into HIV-1 Particles

One of the fascinating and mysterious aspects of Env incorporation is the role of the long CT in particle incorporation. For many years, investigation into Env biology focused almost entirely on features of gp120 and the ectodomain of gp41. Deletion of the CT seemed to have no effect on particle incorporation in the cells often used to produce viruses via transfection, such as 293T or COS, and viruses with a truncated CT could establish spreading infection in the MT-4 T cell line [28]. A seminal work revealing the importance of the CT was published by Murakami and Freed, showing that there are remarkable cell type-specific differences in Env incorporation [13]. They found that incorporation of Env lacking the CT (NL4-3CTdel144-2, abbreviated as CT144) was efficient in 293T and MT-4 cells, intermediate in HeLa cells, and very inefficient in CEM, Jurkat, and primary macrophages. Cells allowing efficient CT144 incorporation were termed permissive for Env incorporation, while those requiring the intact CT were termed nonpermissive (or alternatively, restrictive for CT144 incorporation) [29]. These results suggest that cell-specific factors regulate Env incorporation in a CT-dependent manner. One potential explanation for these findings is the existence of a dominant pathway that depends upon a specific host factor interaction with the CT and determines Env incorporation in the most relevant cells for HIV replication (T cells and macrophages), and excludes Env with a truncated CT (pathway #1 in Figure 2). The very inefficient incorporation of CT144 in nonpermissive cells indicates that a CT-independent pathway, which may act through “passive” incorporation, is blocked in nonpermissive cells. In permissive cells, the CT-dependent pathway is not dominant, and a second pathway (pathway #2 in Figure 2) allows efficient incorporation in the absence of the CT. We envision the CT-dependent pathway in nonpermissive cells as being marked by endocytosis and recycling (Figure 2). However, there may be other explanations that do not invoke trafficking pathways, as the determinants of nonpermissive vs. permissive cell types have not yet been fully defined.

Four potential models have been proposed by Freed and colleagues to explain Env incorporation into HIV particles [30]. These models are somewhat relevant to the study of host factors and pathways involved in Env incorporation. The *passive incorporation model* posits that Env at the PM is passively incorporated into Gag particles that bud from the membrane. The fact that HIV particles acquire a variety of cellular membrane proteins during the formation of particles supports this idea of a promiscuous but nondirected incorporation of Env [13,28,31,32]. Incorporation of Env lacking the long CT occurs in permissive cells, also supporting passive incorporation. However, the exclusion of Env with a truncated CT from nonpermissive cells, including primary T cells and macrophages, argues against this model. The *direct Gag–Env interaction model* proposes that the Env CT directly interacts with Gag to mediate its incorporation into particles. A direct interaction between MA and the CT could explain why deletions or mutations in MA disrupt Env incorporation, as well as why CT truncation can rescue defects in Env incorporation in the presence of MA mutations [31,32,33,34]. Direct binding between MA and CT has been supported by three biochemical studies [35,36,37]. Strong evidence for the interaction between Gag and CT in the virion is also provided by the fact that maturation is required in order for Env to mediate viral fusion, while truncation of the CT relieves the constraint on viral fusion [38,39,40]. Additional supporting evidence comes from studies indicating that trimerization of MA is required for Env incorporation, and that mutations in MA that rescue Env incorporation can also rescue trimerization [41,42]. Together with studies of the MA lattice indicating a hexamer of trimers [35,43], these data suggest that there may be direct contact between the CT and a central aperture in the MA lattice. Recently, cryoET data examining intact virions revealed a different Env positioning, while still supporting a direct interaction between Env and Gag [44]. Although the CT itself was not visualized, Env trimer positioning relative to Gag indicated that the CT lies on the rim of the hexameric Gag-CA lattice rather than over the central aperture. The *Gag–Env cotargeting model* is one where both Gag and Env arrive at a common PM microdomain where particles form, implying that specific host trafficking pathways bring them together. This model fits well with the evidence of nonpermissive and permissive cells for Env incorporation, where distinct host trafficking pathways may regulate CT-dependent Env incorporation. The existence of microdomains for particle assembly is supported most strongly by studies of the lipid composition of HIV-1 virions that indicate enrichment of lipid raft components cholesterol, glycosphingolipids, and phosphoinositides in the viral membrane [45,46]. The cotargeting model fits well with data demonstrating the importance of endocytosis and recycling pathways to Env particle incorporation, as discussed in a subsequent section of this review. Finally, the *indirect Gag–Env interaction model* invokes an adaptor protein that bridges Gag and Env. Proteins binding to the CT have been identified, but at this point there is no evidence for the formation of a ternary complex of Gag–adaptor–Env. We should note that these models are not mutually exclusive. For example, Env and Gag may arrive at a common site of assembly (cotargeting) through the influence of a host factor interacting with the Env CT, and at this site, direct interactions between the CT and the MA lattice may occur. 

## 4. Structure of the CT and Motifs Implicated in Trafficking

The presence of a long CT is a characteristic feature of all lentiviral Env proteins, with the exception of feline immunodeficiency virus and puma lentivirus, in which CTs are 50 and 76 amino acids, respectively [47]. HIV-1, HIV-2, and SIV CTs are similar in length at ~150 amino acids, while the equine infectious anemia virus (EIAV) CT is even longer at ~200 aa [48]. The fact that lentiviruses have evolved long CTs suggests that they must play important roles in viral replication. The importance of the long CT is supported by the finding that an intact HIV-1 CT is required for viral spread in physiologically relevant cell types, as already outlined [13]. This long CT contains multiple functional motifs that promote Env interactions with host cell factors, which will be discussed in later sections of this review. One potential reason for the long CT is that by directing Env into endosomal sorting pathways the virus is limiting cell surface exposure of Env, as a form of immune evasion. Another possible reason that the long CT has evolved is as a means of directing Env trimers into compartments where an as-yet undefined modification to Env occurs that is advantageous for the infectivity of developing virions, while limiting the incorporation of nonfunctional forms of Env. Neither of these explanations is at present completely satisfying. Although many details of Env trafficking remain to be uncovered, host factor interactions with the CT are certainly responsible for Env endocytosis and intracellular trafficking, and likely determine its delivery to the site of particle assembly on the PM.

We next turn to the structure of the CT, in order to highlight its central role in Env trafficking (Figure 3). The CT comprises the last ~150 residues of gp41. The N-terminal part of the CT represents an unstructured loop that contains a highly immunogenic stretch, termed the Kennedy sequence [49,50]. At one time, it was proposed that this sequence was exposed to the extracellular space through a three-membrane-spanning model for gp41 [51], while current topological models place the loop and the rest of the CT inside the cytoplasm [1]. The C-terminal portion of the CT sequence is characterized by a substantial hydrophobic moment, and is traditionally divided into three adjacent regions, termed lentiviral lytic peptides (LLPs) 2, 3, and 1. This terminology comes from early studies indicating that peptide fragments from these three regions form α-helical structures that can bind and destabilize lipid bilayers [52,53,54,55,56]. More recently, an NMR study of a continuous LLP2-3-1 segment of CT that was reconstituted in DPC (dodecylphosphorylcholine) micelles confirmed the α-helical configuration of each LLP, and revealed an extensive hydrophobic membrane-interacting interface for this region of the CT [1]. The majority of aromatic residues within this segment were buried within the lipid of the micelle, and the model derived from these results suggests that each α helix penetrates deeply within the inner leaflet of the viral or cellular membrane bilayer.

A more recent structural view was obtained for the CT fused with the transmembrane domain of gp41 (TMD), which enabled the probing of the CT conformation and quaternary arrangement in the context of a trimer. An NMR model for this construct in DMPC–DHPC (dimyristoylphosphatidylcholine/dihexanoylphosphatiylcholine) bicelles revealed that each protomer of the CT trimer winds around TMD [3]. Within each protomer, the structured regions are arranged in five helical segments (H1–5). H1 helices (amino acids (aa) 741–764, preceding LLP2) directly interact with the TMD and form a trimeric “inner ring”, while H2 helices (aa 769–786 of LLP2) fold around TMD, creating an outer ring [2,3]. The two rings are placed in the polar headgroup region of the lipid bilayer, forming a “baseplate”. Importantly, the H1–H2 segment of the CT spontaneously trimerizes in the presence of TMD, while in the absence of TMD H1–H2 is monomeric [2]. This work emphasizes the central role of TMD in maintaining gp41 trimer conformation. The H3 (residues 790–821 of LLP3) and H4 (aa 826–841 of LLP1) helices also reside in the headgroup region, and are arranged in a hairpin-like manner around the baseplate. At the C-terminus, aa 847–854 of LLP1 form the last short helix (H5). The overall structural organization and stabilization of interhelical interactions within the CT is due to several hydrophobic and polar clusters, rather than selected individual amino acid residues [3]. This structure, along with the functional regions and motifs to be discussed, is depicted in Figure 3A–C. Note that the helices forming the baseplate structure align with the inner layer of the PM, while the unstructured region is depicted as protruding beneath the lipid bilayer of the cell or virion.

A motif that is clearly important in HIV and SIV endocytosis and pathogenesis is the membrane-proximal tyrosine-based motif of the CT. As mentioned, HIV-1 Env was found to be rapidly endocytosed from the PM [11]. A follow-up of this finding mapped the endocytic signal to a critical membrane-proximal tyrosine-based motif on the CT [57]. Similar findings were reported for an SIV mutant, where mutation of a membrane-proximal tyrosine led to greatly increased cell surface levels of Env [58]. For both SIV and HIV, the critical motif was a Yxxø motif, where x represents any amino acid and φ is a residue with a large hydrophobic side chain. Yxxø motifs may interact directly with the μ1 or μ2 subunit of the clathrin adaptor complexes AP-1 and AP-2, respectively; this was subsequently proven by several groups for this CT motif [12,59,60]. Indeed, membrane-proximal Yxxø motifs are found in CTs of almost all retroviruses and, like HIV-1 Y_712_SPL, are involved in clathrin- and AP-2-mediated endocytosis of Env from the PM [59,60,61]. This aspect of the lifecycle will be further discussed in the section on endocytosis below. Another remarkable finding related to the membrane-proximal Yxxø motif came from studies of HIV-1 particle formation in polarized cells. Lodge and coworkers reported that in MDCK cells, virus release was polarized to the basolateral surface in an Env-dependent manner, and that the polarization was dependent upon MA and the Env CT [62]. Basolateral targeting of Env was abolished upon mutation of the membrane-proximal tyrosine, implicating this motif in a distinct aspect of intracellular trafficking [63]. Another compelling story implicating the membrane-proximal Yxxø motif in lentiviral pathogenesis comes from studies of SIVmac239 engineered to be deficient in this motif. A virus deleted for GY_720_ within the membrane-proximal GYRPV sequence, termed ΔGY, causes acute infection in pigtail macaques in a manner equivalent to the wildtype virus, but the macaques subsequently control the infection, and there is no depletion of CD4+ T cells in the blood or gut, as occurs with wildtype SIVmac239 [64]. When following up this intriguing finding connecting a trafficking signal to pathogenesis, Lawrence and coworkers performed a detailed study examining viruses from animals that progressed to AIDS following ΔGY infection [65]. Remarkably, disease progression was linked to novel amino acid changes arising within the CT that recreated signals leading to endocytosis and polarized sorting of Env. In some cases, a small deletion recreated a Yxxø motif at a site distal to the disrupted GYRPV motif, restoring both endocytosis and basolateral sorting. In one animal, a novel mutation leading to pathogenesis restored polarized sorting but not endocytosis, suggesting that the basolateral sorting conferred by trafficking motifs may be of particular importance in viral replication and pathogenesis. We note that HIV and SIV do not infect polarized epithelial cells, and that T cells and macrophages do not have basolateral and apical surfaces that are strictly equivalent to those in polarized cells. The implication of these studies, however, is that trafficking to specific membrane microdomains within infected T cells and macrophages requires this motif, and is important in HIV replication and pathogenesis. 

Moving toward the C-terminus of CT, there are two “inhibitory sequences” (ISs) that span the H1–H2 region: IS1 (aa 748–761) and IS2 (aa 762–785). These sequences were designated as inhibitory in a study utilizing deletional mutagenesis and Env CT chimeras, where elimination of one or both ISs led to an increase in cell surface expression of Env [66]. Note that IS2 in this study overlaps considerably with LLP2 (see Figure 3). Both sequences were found to act as Golgi retrieval signals, and the presence of either IS1 or IS2 was sufficient to reduce surface Env expression. Although Yxxø and dileucine motifs are present in IS2, the inhibition of Env surface expression by this particular sequence does not seem to depend on interaction with AP-2, as the mutation of Yxxø and dileucines in this region failed to shift Env to the PM [66]. More recently, these inhibitory sequences were shown to be important for CT interaction with the retromer complex, a component of the endosomal sorting machinery that regulates cargo trafficking from the endosome to the Golgi apparatus [67,68]. Direct binding of the CT to retromer components Vps35 and Vps26 was demonstrated, and depletion of retromer led to a shift in Env from intracellular locations to the PM. Together, these studies identify the region represented by IS1 and IS2 to be important for intracellular trafficking of Env, moving Env from endosomal compartments to the Golgi apparatus. 

In addition to conserved tyrosine and dileucine motifs in the CT, there are conserved tryptophans that may play a role in viral replication. A recent study focused on the role of two of these conserved residues, W_757_ and W_790_, in cell-to-cell transmission [69]. The W_757_ virus exhibited a significant defect in fusion and infectivity. Moreover, this virus exhibited a defect in Env polarization to sites of cell–cell contact and in cell–cell spread, suggesting that W_757_ within the LLP2/H2 helix plays a particularly important role in cell–cell transmission events. 

LLP3 appears to contain critical determinants for Env incorporation into HIV-1 particles. Murakami and Freed demonstrated that small deletions in this region elicited Env incorporation defects along with defects in replication [70]. Revertant viruses were derived that rescued these defects through the creation of a mutation in MA (V_34_I). A comprehensive mutagenesis study of tyrosine and dileucine motifs in the CT concluded that the nine-residue stretch at the N-terminus of LLP3 (Y_795_WWNLLQYW_802_) plays a particularly important role in HIV replication in T cells [71]. The YW_795_ motif was shown to be critical for the engagement of Rab11-FIP1C and trafficking of Env through the endosomal recycling compartment (ERC) for subsequent incorporation into the particles [72,73]. The role of Rab11-FIP1C and the ERC will be discussed in detail in a subsequent section of this review. Similarly, the disruption of a second nearby YW motif in LLP3 (YW_802_) resulted in reduced Env particle incorporation and reduced fusogenicity [71,74,75]. In one study, this YW_802_ was implicated in the interaction of the CT with TIP47 (also known as perilipin 3), which promoted retrograde transport from endosome to TGN [76]. However, subsequent work did not substantiate previous findings, either by overexpressing or silencing TIP47 [77]. Another study examining YW_802_ and LL_799_ mutations within this segment found that cell-free infectivity was significantly impaired, while cell–cell transmission was largely unaffected [78]. LL_799_ was implicated in binding to prohibitin 1/2 in a separate study [79]. Although the precise nature of prohibitin involvement in Env trafficking remains unclear, preventing CT–prohibitin interaction abrogated HIV replication in T cells [79]. The replication defect of the LL_799_ mutant was also observed in another study, while mutation of the second dileucine in LLP3 (LL_814_) resulted in only a mild reduction in Env fusogenicity [71]. 

The LLP1 region of CT contains a C-terminal LL_855_ motif that is important in Env trafficking and particle incorporation. This motif, together with the membrane-proximal Yxxø motif, has been implicated in clathrin-dependent endocytosis mediated by AP-2 [61]. Another report, however, linked LL_855_ with AP-1 binding rather than AP-2, and demonstrated that mutation of these residues shifted the intracellular localization of Env, while having little effect on the rate of endocytosis [80]. LLP1 also incorporates an abundance of arginine residues [56]. In fact, arginines are specifically “preferred” in the CT of HIV Env, unlike gp120 or the ectodomain of gp41, where they can be functionally exchanged for lysines [56,81]. Although mutations of arginines in the CT to lysines did not result in global structural perturbations to the Env, as judged by antibody neutralization profiles, they did reduce Env particle incorporation and fusogenicity, thus affecting the replication kinetics of the mutant viruses. The most prominent effects were observed for R to K substitutions in the C-terminal H4–H5 segment [81]. 

## 5. Env Protein Synthesis and Trafficking Part 1: Secretory Pathway to PM

This section describes in more detail the synthesis and trafficking of Env through the secretory pathway to the PM. HIV-1 Env is encoded on a bicistronic, singly spliced RNA transcript that encodes the *vpu* open reading frame (ORF) followed by the *env* ORF. Translation of Env occurs via a leaky ribosome scanning mechanism, in which the ribosome bypasses the *vpu* AUG due to its weak Kozak context [82,83]. This can be important in regulating the degree of Env synthesis, as improvements in the Kozak context of the *vpu* start codon can result in diminished Env translation, while loss of the *vpu* start codon leads to enhanced synthesis of Env [84,85,86]. Env is synthesized as a precursor glycoprotein (gp160) on the rough endoplasmic reticulum (RER). The precursor gp160 includes a signal peptide (SP) at its N-terminus that mediates translocation into the ER lumen, and is subsequently removed by ER signal peptidases. The SP of HIV-1 itself has been noted to have some unusual features. The expression of Env in some systems has been noted to be quite inefficient, and can be enhanced by the replacement of the HIV-1 SP with SPs derived from alternative glycoproteins [87,88]. Exchange of the SP in the proviral context, however, did not lead to more effective incorporation of Env into virions, or to enhanced replication in mammalian cells [89]. The Env SP contains a positively charged N-terminal region, a hydrophobic region, and a cleavage domain, typical of other SPs. It has been reported that transmitted HIV-1 isolates have an overrepresentation of basic amino acids in the N-terminal region of the SP that is lost in chronic phase isolates [90,91]. The basic signature led to a higher proportion of unprocessed oligomannose residues on gp120 compared with the glycan profile of Env proteins from isolates lacking this signature. The proportion of high mannose vs. complex glycan modifications can be affected by the rapidity of transit through the secretory pathway, or by steric constraints that limit the access of ER and Golgi α-mannosidases to the oligomannose glycans [92]. Both glycosylation and antigenicity were significantly impacted by these differences in the SP [93]. The impact of signal peptide sequence diversity on Env glycan content can have important effects on virus neutralization [94,95].

Gp160 is glycosylated in the ER, with a predominance of *N*-linked glycans and fewer *O*-linked glycans [96,97]. The gp160 monomer is very heavily glycosylated, with 26–30 *N*-linked glycosylation sites [98,99]. The folding of gp160 is assisted by chaperones in the ER, including calnexin, calreticulin, and GRP78-BiP [100,101,102]. Calnexin plays a role in the formation of a network of intramolecular disulfide bonds in conjunction with oxidoreductases [103]. The disulfide linkages include nine within gp120, and one in gp41; mutations of cysteines that form these bonds in general lead to misfolding and impaired function of Env, with a few exceptions [104]. Multimerization occurs in the ER, with a predominance of trimers formed, although some biochemical evidence for tetramers has also been shown [105,106]. Clearly, the major relevant oligomer is the trimer, as this is the dominant form found on virions when examined by cryoET [107]. Env trimers are equipped for transit to the Golgi apparatus, where additional modifications to Env take place. The high-mannose oligosaccharides on gp160 are modified by the removal of some mannose residues by α-mannosidases, followed by the addition of many additional sugars to create complex oligosaccharide chains. There is heterogeneity in the glycosylation process, including in the reactions leading to the formation of complex glycans; consequently, a significant population of unprocessed oligomannose glycans remains on the surface of gp120 in a cluster termed the intrinsic mannose patch [108]. Cleavage of the gp160 monomers of each homotrimer into gp120 (SU) and gp41 (TM) occurs in the Golgi apparatus through the action of furin-like proteases [109,110]. Following cleavage, gp120 and gp41 remain associated through noncovalent interactions. Cleavage of gp160 is essential for particle infectivity, as it frees the N-terminal fusion peptide domain of gp41 to allow fusion following receptor and coreceptor binding on a target cell. The cleaved Env trimers are subsequently delivered to the PM through vesicular transport from the TGN. 

A subpopulation of HIV-1 Env on the surface of cells remains uncleaved. Three populations of Env on the cell surface were described by Zhang and coworkers: cleaved Env modified by complex glycans; a small population of uncleaved Env modified by complex glycans; and uncleaved Env that lacked complex glycan modification [111]. This last group of uncleaved oligomers was recognized by poorly neutralizing antibodies, and reached the cell surface even in the presence of brefeldin A, suggesting the existence of a second pathway for Env trimers to reach the PM that bypasses the Golgi apparatus. However, the uncleaved Env was largely excluded from virion incorporation, potentially serving as an immunologic decoy with distinct antigenic properties from virion-incorporated, fully cleaved Env [112,113]. 

Host factors inhibiting viral replication can provide insights into specific steps in the lifecycle. Guanylate binding protein 5 (GBP5) is one such factor that acts on Env as it travels through the secretory pathway. GBP5 was identified in a genetic screen for interferon-responsive host factors that inhibit HIV replication [114]. GBP5 is a GTPase localized to the Golgi apparatus that inhibits *N*-linked glycosylation, disrupts gp160 cleavage, and impairs the incorporation of functional Env trimers into particles [115]. The effects of GBP5 on HIV-1 replication were particularly pronounced in macrophages, and there was significant donor-to-donor variability in this phenotype. Remarkably, deletion of *vpu* allowed HIV-1 to overcome the inhibition introduced by GBP5 through enhanced translation of Env. This suggests that in some situations or anatomical compartments, it may be advantageous for the virus to lose Vpu expression and produce more Env. Inhibition of furin-mediated protease activity in the TGN was shown to be the major mechanism of action of GBP5 (and GBP2), along with evidence that GBP5 can inhibit other viruses that rely on cleavage of their glycoproteins by furin [97]. 

## 6. Env Protein Trafficking Part 2: Endocytosis

It is logical to assume that once Env has reached the PM, the simplest route for incorporation would be to remain on the PM and pair up with Gag for incorporation into developing particles. Instead, however, after reaching the PM, HIV-1 Env is rapidly endocytosed [11,12,57]. Endocytosis is mediated by a membrane-proximal tyrosine-based motif in the Env CT (Yxxφ). As described in the section on CT structure and motifs, Yxxφ motifs direct the internalization of cellular transmembrane proteins by interacting with the μ subunits of adaptor protein complexes, in this case μ2 of the AP-2 complex [12,59,60]. Substitution of the critical tyrosine residue (Y_712_) within this motif led to greatly reduced rates of endocytosis and accumulation of Env on the PM for both HIV-1 and simian immunodeficiency virus (SIV) [57,58,60,116]. For HIV-1 Env, an additional Yxxφ motif is present at a more distal position from the membrane (Y_763_), but mutagenesis of this motif did not reveal a role in the rapid endocytosis of Env [57]. In addition to the membrane-proximal tyrosine-based motif, the C-terminal dileucine of the CT was identified as mediating AP-2-dependent endocytosis [61]. This study found that both signals act to mediate endocytosis independently, and that Env remains on the cell surface only if both signals are mutated. There thus appears to be functional redundancy in the CT for endocytosis. The membrane-proximal Yxxφ motif can bind to AP-1 in addition to AP-2, and this may have consequences for subsequent trafficking events within the endosomal system [59,60]. 

What is the fate of Env following initial endocytosis? The sequence of events that occur following endocytosis is at present poorly understood. If we use endocytosis of activated EGFR as a model, AP-2 and clathrin-mediated endocytosis first leads to the Rab5+ early endosome (EE). From the EE, several routes are possible [117]. Under some conditions, EGFR is delivered to the Rab7+ late endosome/lysosome for degradation. A second route, most commonly taken by moderately activated receptors, is recycling through the Rab11+ endosomal recycling compartment (ERC). While Env is not EGFR, there is evidence that Env traffics through early endosomes, the ERC, and to lysosomes for degradation [73,118]. Further complicating the picture is evidence that Env may traffic from endosomes to the Golgi apparatus through interactions with the retromer complex [67]. The recycling of Env to the site of assembly for particle incorporation is an area of active investigation [72,73,119]. The evidence for specific trafficking of Env within endosomal pathways will next be described in more detail.

## 7. Env Protein Trafficking Part 3: Host Factors Involved in Endosomal Sorting/Recycling

Multiple trafficking motifs within the CT as well as potential interacting factors have been described in the sections above. Here we will focus on host factors involved in recycling, and attempt to derive a coherent model for Env trafficking events that occur following endocytosis of the trimer from the PM. We note that the importance of endosomal recycling to particle incorporation is currently supported by incomplete evidence, and other trafficking and incorporation pathways for Env can be proposed. Nevertheless, we believe this model is useful for explaining existing data regarding endocytic motifs in the CT and the role of specific host recycling factors. This model should serve as a framework for ongoing investigation into Env trafficking, and not be regarded as final. 

The Rab11 family of proteins play central roles in intracellular trafficking and recycling of host cell membrane proteins [120]. They localize to the TGN and post-Golgi vesicles, and are also concentrated in the ERC, where they control trafficking and recycling to the PM. Rab11 family members form complexes with motor proteins, and interactions with cellular cargo are mediated by Rab11 family interacting proteins (Rab11-FIPs) [120]. Our group investigated the role of dominant negative S_25_N mutant of Rab11a and a constitutively active Rab11a S_20_V mutant some years ago, testing the hypothesis that inhibition of Rab11a-mediated trafficking may alter Env trafficking. While there was little effect of the dominant negative S_25_N Rab11a on Env incorporation into particles, overexpression of the active GTP-bound form caused a reduction in Env incorporation. Because this form of Rab11a binds avidly to Rab11-FIPs, potentially depleting these trafficking adaptors upon overexpression, we undertook an investigation of each of the Rab11-FIP family members. Depletion of each family member individually led us to the finding that Rab11-FIP1C (FIP1C) was involved in Env trafficking and particle incorporation [119]. Further investigation revealed that FIP1C interactions with Rab14, rather than Rab11, were responsible for mediating Env incorporation through interactions with FIP1C. Importantly, the effects of FIP1C on trafficking were dependent upon an intact CT. The significance of this finding is that specific cellular recycling pathways direct Env to the PM for incorporation into developing particles, and interactions with cellular recycling pathways are dependent upon signals in the CT. 

In follow-up studies, we mapped the putative interacting region on the CT using a series of CT mutants and an assay whereby GFP-FIP1C is relocalized to the periphery of cells upon expression of Env bearing an intact CT, but not Env with CT144. A YW substitution mutant in the proximal portion of LLP3, YW_795_/SL, failed to redirect to the cell periphery upon Env expression [72]. Remarkably, this mutant was defective for Env particle incorporation in nonpermissive cell lines (H9, Jurkat, primary MDM), but was not defective for Env incorporation in permissive cell lines such as MT-4 or 293T. Replication of YW_795_/SL was very poor in H9 cells. However, after months of passage, a revertant virus was derived that restored Env incorporation and replication to wildtype levels. This revertant maintained the YW_795_/SL substitution but introduced a second-site mutation near the C-terminus of the CT, L_850_/S. The mechanism for restoration of Env incorporation by this second-site mutation in the CT remains unknown. These results are significant because they connect this altered YW motif to cell type-dependent particle incorporation, suggesting that this motif, or this proximal region of LLP3, is key to understanding Env particle incorporation. FIP1C was suggested as a potential binding partner for this region, although to date there is no evidence of direct binding. Another study employed the C-terminal Rab-binding domain of FIP1C (FIP1C_560-649_) to further investigate the role of the ERC in Env trafficking [71]. Expression of this fragment acted in a dominant negative manner to trap Env within the perinuclear ERC in a CT-dependent fashion. Mutation of the membrane-proximal Y_712_SPL or YW_795_ motif prevented trapping in the ERC, providing evidence that endocytosis leads to the appearance of Env in the ERC, and suggesting that the YW_795_ motif is critical for trafficking through the ERC. 

How do these findings supporting a role for active CT-dependent recycling of Env fit with all of the other CT motif and binding partner data discussed in this review? Clearly there are still many unknown aspects of Env trafficking. However, we can conclude that there are multiple paths that can be taken by Env following endocytosis from the PM. The major pathways are likely ERC-mediated recycling to the PM, where Env may be cotargeted to Gag microdomains; retromer-mediated trafficking to the Golgi apparatus; and a default pathway of early-to-late endosome maturation followed by fusion with lysosomes and lysosomal degradation (outlined in Figure 4). One area of uncertainty is the role of AP-1 in Env trafficking. As described above, AP-1 can bind to the membrane-proximal Yxxφ motif or the C-terminal dileucine motif of the CT. AP-1 is associated with both TGN and endosomes. Knockdown of AP-1A causes a defect in basolateral sorting in epithelial cells [121]. Given the previously mentioned results indicating that the Env CT membrane-proximal Yxxφ motif is required for basolateral sorting and the intriguing results that this sorting mediates SIVmac239 replication and pathogenesis in a macaque model [63,65], it is tempting to speculate that AP-1 interactions with the CT may play an as-yet undefined role in trafficking of Env to the particle assembly site. In Figure 4, this is depicted as movement of Env from the TGN to the site of assembly. However, the AP-1 complex is also associated with recycling endosomes [122,123,124]; thus, it is possible that factors regulating outward trafficking from the ERC, such as FIP1C, may intersect with AP-1-mediated vesicular trafficking to the site of productive assembly.

## 8. Areas for Future Investigation

There are many aspects of Env trafficking that remain to be resolved. Among them are the following:

*Identify the molecular basis for cell type-specific incorporation of Env.* Although we have some clues that this may be tied to interactions with the proximal portion of LLP3, and that recycling factors appear to be involved, this area needs further investigation. We have outlined a role for FIP1C above. However, there is some redundancy in Rab-dependent recycling pathways, so perhaps the involvement of FIP1C is best viewed as a window into an important role for outward sorting from the ERC. Defining the factor(s) that account for CT-dependent vs. CT-independent incorporation into particles is very likely to provide major clues to the nature of the particle assembly site itself. 

*Define the role of the AP-1 complex in Env trafficking and particle incorporation.* As discussed above, basolateral targeting appears to be an important role of the membrane-proximal Yxxφ motif. Studies aiming to understand where the directional AP-1-mediated trafficking of Env occurs, performed in both nonpermissive and permissive cell types, may be very informative. Although HIV-1 does not replicate in polarized epithelial cells in vivo, the pathways and host factor interactions identified in these cells may be relevant to replication in T cells and macrophages. Instead of the basolateral targeting seen in epithelial cells, the same processes in infected T cells or macrophages in vivo might direct Env to a specific PM microdomain.

*Further dissection of the role of recycling endosomes in Env incorporation.* FIP1C is a clue to the importance of recycling compartments in studies of Env incorporation. Additional investigation into the role of FIP1C and other FIPs in Env trafficking is needed. For example, it remains unclear whether there is a direct interaction of FIP1C with motifs in the LLP3 region of the CT. Recycling endosomes are not limited to the pericentriolar region, but instead form a network of tubular recycling endosomes that is critical to the sorting of many transmembrane glycoproteins [125,126]. Studies are underway to examine the role of tubular recycling endosomes and the host factors that regulate their function in Env trafficking and particle incorporation.

*Define the complete sequence of events in Env trafficking following endocytosis.* It will be important to understand the sequential trafficking of Env in the cell. Such studies are complicated by the presence of Env within multiple compartments in an infected cell: ER, Golgi apparatus, PM, ERC, lysosome, and vesicles will all be found to have a population of Env. To follow Env sequentially, novel approaches are needed for pulse labeling and imaging Env from its initial appearance at the PM and tracking its movement within endosomal pathways following endocytosis. Using a robust pulse label of Env, and utilizing specific CT mutants along with the depletion of individual components of the pathways presented in Figure 4, we should be able to define each sequential step of the Env trafficking pathway leading to particle incorporation. 

## 9. Conclusions 

Env trafficking is regulated by the gp41 CT and its interactions with host trafficking factors. There are well-described interactions of some motifs in the CT with known trafficking proteins such as AP-1, AP-2, and retromer; yet the sequence of events leading to particle incorporation remains incompletely understood. Env is found in multiple endocytic compartments following endocytosis from the PM. Multiple host trafficking molecules are likely to take part in endocytic sorting events, leading to retrograde transport to the Golgi apparatus, degradation in the lysosome, or outward sorting to the PM. An area of continued interest is the role of the recycling factors involved in Env incorporation into particles. Recycling pathways appear to hold the key to understanding cell type-specific incorporation of Env.

## Figures and Tables

**Figure 1 viruses-14-01729-f001:**
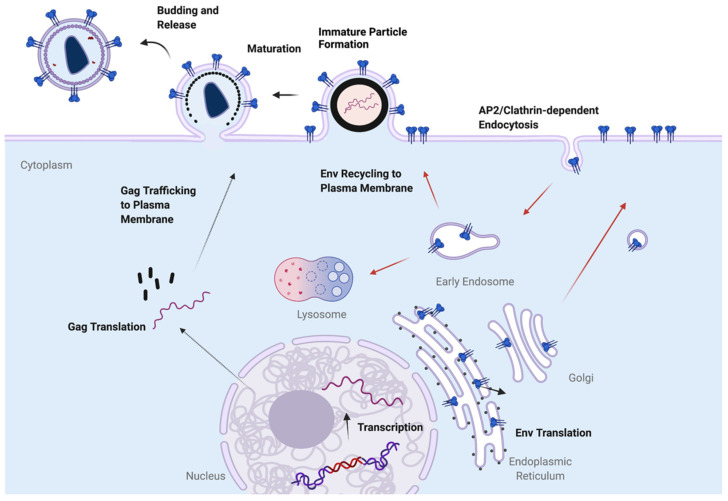
Overview of HIV-1 assembly process. Gag is translated on cytosolic ribosomes and trafficked to the PM. Env is synthesized on the rough ER and transits the secretory pathway to reach the PM. Following arrival at the PM, Env is rapidly endocytosed. Env may follow several routes after arriving in the early endosomal compartment, including proceeding to late endosomes and the lysosome for degradation, or recycling to the PM to pair up with Gag for incorporation into developing particles. Note that this is a simplified schematic, and thus does not indicate all possible pathways for Env trafficking.

**Figure 2 viruses-14-01729-f002:**
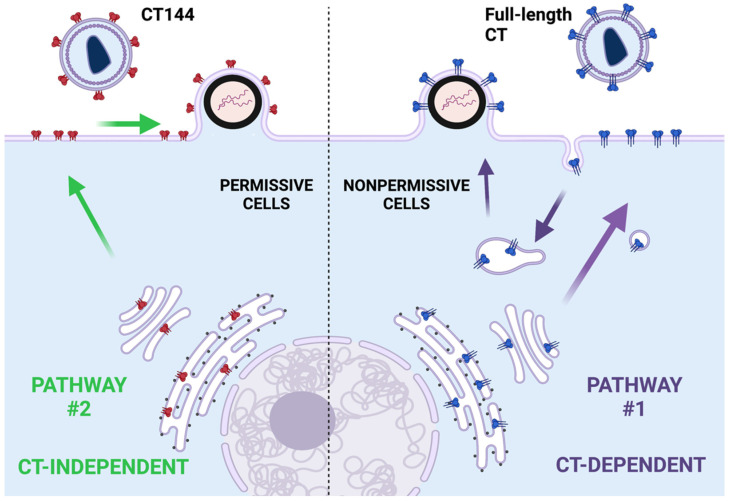
Schematic diagram outlining the existence of different pathways for Env available in nonpermissive vs. permissive cells. Nonpermissive cells exhibit a dominant CT-dependent pathway for incorporation of Env, while permissive cells allow either full-length CT or CT144 (truncated CT) Env to be incorporated. The CT-independent pathway is depicted as passive incorporation at the PM. The CT-independent pathway is lacking or blocked in nonpermissive cells.

**Figure 3 viruses-14-01729-f003:**
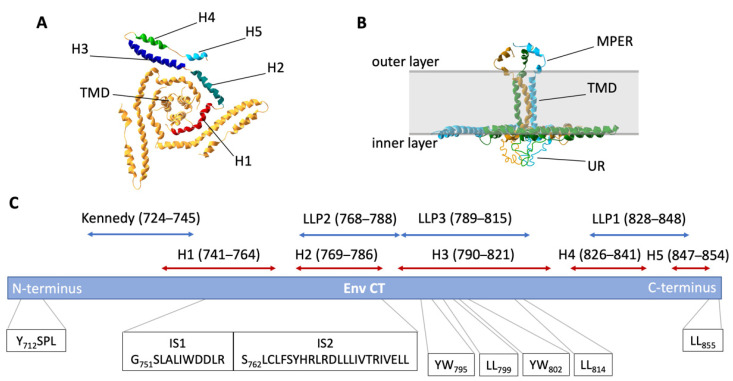
(**A**) A structural model of trimeric CT with colored α helices labeled as H1–H5; from PDB 7LOH [3]. TMD = transmembrane domain. (**B**) Side view of trimeric TMD/CT; from PDB 7LOI [3]. MPER = membrane-proximal ectodomain region; UR = unstructured region. Rendering of structures was performed with Swiss PDB Viewer. (**C**) Linear schematic of Env CT, showing relative positions of domains and motifs discussed in the text. Numbering is based on HXB2.

**Figure 4 viruses-14-01729-f004:**
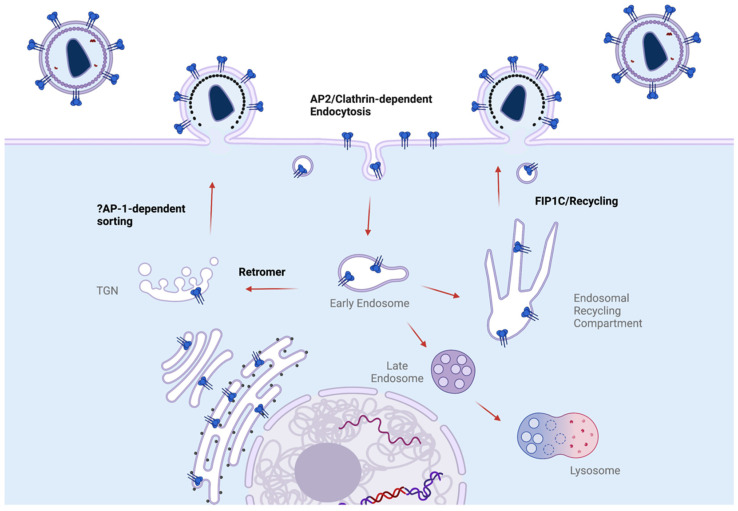
Schematic of pathways available for Env trafficking from the early endosome. Env may follow a pathway to degradation through early-to-late endosomes, which fuse with the lysosome. Two potentially productive pathways for Env trafficking back to the surface are shown. The endosomal recycling compartment has been shown to play a role in Env incorporation, mediating recycling to the PM. The retromer complex directs Env to the TGN, where AP-1-dependent sorting may play a role in particle incorporation of Env.

## Data Availability

Not applicable.

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
