# Peer review of "Viral and Host Factors Regulating HIV-1 Envelope Protein Trafficking and Particle Incorporation"

_viruses, 2022, doi:10.3390/v14081729_

Round 1
Reviewer 1 Report
This is a very well-written and nicely organized review on everything you always wanted to know about HIV-1 Env. The authors deserve praise for their heroic effort to condense more than 30 years of research on HIV-1 Env into a comprehensive, yet compact compendium that will serve as a reference guide for future generations of HIV researchers. I am not an expert on HIV-1 Env and reading this manuscript made me realize just how much we still don’t know about the mechanisms of Env incorporation into viral particles. Overall, this is a fantastic piece of work. However, being an Env non-aficionado, I have a list of comments that the authors may want to consider.
- In my opinion, the review puts a significant focus on discussing structural features of Env and less focus on host factor contributions. As such, the authors might want to consider inclusion of “Structural” in their title.
- One of the most interesting insights for me was the unusual way of Env trafficking. The authors state that Env first traffics to the cell surface, gets cleaved in the trans-golgi compartment on the way, but then gets rapidly internalized and recycled to the cell surface. I find the evidence that this happens very convincing. What I did not completely understand is what is the experimental evidence that virus-incorporated Env is derived from the pool of recycled Env, rather than from residual Env that did not get internalized. Maybe the authors can make that clearer. I would argue that the rapid internalization of Env helps the virus limit the number of Env molecules packaged into a virion. As I said, I am not an expert on Env and I would find it helpful if the authors could explain the evidence to support the notion that Env packaged into a virion comes from the pool of recycled Env.
- Another point that may be of interest to Env non-aficionados is that not all Env appears to be processed in the trans-golgi, especially when Env is transiently over-expressed. Is there yet another pathway that allows Env to bypass the cleavage machinery in the trans-golgi; or is this simply a saturation phenomenon (and thus an experimental artifact)?
- Beginning with line 147, the authors discuss four potential models for Env incorporation. Can they comment on which of these models would require/not require prior endocytosis/recycling of Env?
- The most challenging chapter for me to read was chapter 4 (Structure of the CT and motifs implicated in trafficking), which starts with line 186. The authors include figure 3 but they do not refer to that figure until the end of the third paragraph (line 240). Reference to figure 3 should be much sooner. Also, they should integrate some of the elements discussed in figure 3C into figure 3B. For instance, they state that H1 helices directly interact with the TMD and form a trimeric inner ring (lines 228-230). To me, figure 3B implies that the Env CT aligns itself along the inner membrane layer. It is hard for me to understand how the three spread-out CT monomers could form a trimer and still be flat against the inner layer of the PM. Maybe adding labels for H1-H2 in figure 3B would make that clearer.
- Lines 268-274: The authors talk about the involvement of the membrane proximal tyrosine motif for the basolateral sorting of Env. Can the authors comment on the physiological significance of this observation? Is there a basolateral compartment in T cells? What about macrophages? The authors mention that in macrophages, budding occurs on internal membranes that connect to the cell surface. But those are not basolateral membranes, right?
- Line 275: Can the authors clarify the meaning of “inhibitory sequences”? They note that this refers to reduced Env surface expression; but does it also affect viral infectivity?
- Lines 351-353:” The basic signature led to a higher proportion of high mannose residues on gp120 compared with the glycan profile of Env proteins from isolates lacking this signature”. Does that mean these Env molecules residue longer in the ER (i.e., are trafficking impaired)?
- Can the authors comment/speculate on why Env forms trimers (rather than dimers or tetramers, [or hexamers like Gag])? The authors state that trimerization occurs already in the ER, which rules out Gag as a “Chaperone”.
Minor:
- Lines 33-35: The authors are very specific about copies of HIV-1 Env and SARS-CoV-2 spikes in a virion but are vague about number of Env proteins in Influenza, Measles, Ebola, and RSV virus. Is this because numbers are unavailable for these viruses?
- Lines 82-83: “Complete cleavage of Gag is essential for maturation”. Is that true? What is the evidence that particles containing small amounts of residual P55gag are non-infectious?
- Can the authors define “Retromer”?
Reviewer 2 Report
The piece is an excellent and thought-provoking minireview of Env structure, function, and trafficking. I particularly liked the broad view of the topic. The last section seems more like a presentation of the lab’s own findings and may be of less interest to others than to the authors themselves, but that’s fine given the nature of the article. I have a few minor suggestions for improvement.
1. On page 2 the authors introduce the terms “permissive” and “nonpermissive” but then use “restrictive” instead of “nonpermissive” throughout the manuscript. This could be confusing to some readers, and “restrictive” tends to connote a host restriction or tropism type of phenotype in our field. I favor using “nonpermissive” to describe a cell-dependent phenotype of a mutant.
2. Has the above-mentioned phenotype never been tested via heterokaryon analysis? This might be worth stating.
3. Bottom of page 4: in the context of direct binding between MA and CT, consider citing PMID: 11000206
4. Top of page 5: in the context of MA arrangement, the authors should include PMID: 34353956
5. Page 5: a diagram depicting the models for Env incorporation could be helpful.
6. Since this is a review, I would encourage the authors to speculate as to why lentiviral TM proteins have evolved to have long cytoplasmic tails while other retroviruses do not require them.
